# Searching for the Mechanism of Action of Extremely Low Frequency Electromagnetic Field—The Pilot fNIRS Research

**DOI:** 10.3390/ijerph19074012

**Published:** 2022-03-28

**Authors:** Karolina Jezierska, Anna Sękowska-Namiotko, Bartłomiej Pala, Danuta Lietz-Kijak, Helena Gronwald, Wojciech Podraza

**Affiliations:** 1Department of Medical Physics, Pomeranian Medical University, 71-073 Szczecin, Poland; karolina.jezierska@pum.edu.pl (K.J.); annsekow@student.pg.edu.pl (A.S.-N.); pala.b@edu.pum.edu.pl (B.P.); 2Department of Propaedeutic, Physical Diagnostics and Dental Physiotherapy, Pomeranian Medical University, 70-204 Szczecin, Poland; danuta.lietz.kijak@pum.edu.pl (D.L.-K.); helena.gronwald@pum.edu.pl (H.G.)

**Keywords:** magnetic stimulation physiotherapy, neurodegenerative disease, haemodynamic response, functional near-infrared spectroscopy

## Abstract

There is an ongoing debate on the benefits of magnetic stimulation in neurological disorders. Objectives: We aimed to evaluate the influence of magnetic stimulation on blood oxygenation of the motor cortex using functional near-infrared spectroscopy (fNIRS). Methods: A total of 16 healthy volunteer participants were subjected to four protocols. In the first two protocols, the participants remained at rest without (and then with) magnetic stimulation. In the next two protocols, motor cortex stimulation was achieved using a finger-tapping task, with and without magnetic stimulation. Changes in blood oxygenation levels within the motor cortex were recorded and analysed. Results: No characteristic changes in the blood oxygenation level-dependent responses were observed in resting participants after magnetic stimulation. No statistically significant difference was observed in the amplitude of the fNIRS signal before and after magnetic stimulation. We observed characteristic blood oxygenation level-dependent responses after the finger-tapping task in the second protocol, but not after magnetic stimulation. Conclusions: Although we did not observe any measurable effect of the magnetic field on the haemodynamic response of the motor cortex, understanding the mechanism(s) of magnetic stimulation may be important. Additional, detailed studies are needed to prove or negate the potential of this medical procedure.

## 1. Introduction

Blood supply is essential for proper central nervous system functioning [1]. In our body, the brain is the most metabolically active organ. Many factors affect the brain’s functioning, with the most important being the availability of oxygen and glucose. Both oxygen and glucose are necessary for oxidative cellular respiration and subsequent energy production in the form of adenosine triphosphate (ATP) molecules. ATP is the energy source for most intracellular anabolic processes [2]. The brain cortex is divided into many different functioning areas, as described by Korbinian Brodmann in the 20th century [3]. Each area possesses certain functions, and area-specific activation changes with the action levels of the human body. During neural activity, the delivery of oxygenated hemoglobin increases locally according to an individual area’s metabolic demand. A rise in the oxygen consumption of any one brain area will increase local cerebral blood flow. This increases the concentration of oxyhemoglobin (oxy-Hb) while reducing deoxyhemoglobin (deoxy-Hb) levels. This process is known as a hemodynamic response, neurovascular coupling, or ‘functional hyperemia’, and is quantified using functional near-infrared spectroscopy (fNIRS) [4,5,6].

Oxy-Hb and deoxy-Hb absorb different wavelengths of near-infrared light. As such, the channel network of photon emitters and detectors in fNIRS allows for measuring fluctuations in the levels of these two types of hemoglobin in the blood in any given area. These fluctuations can be quantified using the modified Beer-Lambert law [7,8,9]. Previously, fNIRS was used to detect the seizure source in epilepsy [10], evaluate the effects of deep brain stimulation in patients with Parkinson’s disease [11], and assess brain responses in completely locked-in patients with amyotrophic lateral sclerosis (ALS) [12]. fNIRS can also be applied to the developmental sciences, including cognitive neuroscience [5,13,14]. Nonetheless, the applicability of fNIRS has been limited to scientific research, despite multiple reports comprehensively describing its reliability and providing novel methods for analyzing the resultant data [5,9,13,14].

Not only fNIRS allows to measure neurovascular coupling function. Functional Magnetic Resonance Imaging (fMRI) is the second entirely different method which tests the same phenomenon. The final effects of mentioned hemodynamic response are changes in oxy-Hb and deoxy-Hb levels. Magnetic properties of oxy-Hb are similar to surrounding tissues whereas those of deoxy-Hb are very different, since in contrast to weakly diamagnetic oxy-Hb, deoxy-Hb is significantly paramagnetic. Replacement of deoxygenated blood by oxygenated blood changes local magnetic environment which can be visualized by this method. fMRI reflects the blood oxygen level-dependent (BOLD) changes in the MRI signal. Both fNIRS and fMRI map brain activity by measuring changes in oxy-hemoglobin and deoxy- hemoglobin concentrations. Many research projects using both fNIRS and fMRI investigate the metabolic response of specific brain structures to a specific stimulus (e.g., visual, auditory, gustatory) or task (e.g., motor—finger tapping). There are the advantages and disadvantages of both methods. fNIRS is not as sensitive to movement as fMRI. The fNIRS test can be performed on patients while walking, whereas it is impossible for fMRI, where subjects must remain still. fNIRS offers more multimodal measurements compare to fMRI [e.g., fNIRS and fMRI, fNIRS and electroencephalography (EEG), fNIRS and positron emission tomography (PET)]. fMRI is characterized by high spatial resolution, measures both brain function and structure. fNIRS has a poor spatial resolution and low signal-to-noise ratio. In turn, fMRI has relatively poor temporal resolution, fNIRS moderate one compared to highest for EEG. Other disadvantages of fMRI are: very loud noise when scanning, limited examination space, which may prevent people with claustrophobia from being examined. Finally, the relatively low cost of fNIRS, the portability of the equipment is an advantage over expensive and immobile fMRI [15].

It should be emphasized that despite such an advanced technique of magnetic resonance imaging, it is completely useless for the described research project. The tested factor is the electromagnetic field, the strength of which is several orders of magnitude smaller than that generated by the fMRI device. The interference of both fields makes it impossible to study one of them.

The human body is exposed to various types of magnetic and electric fields from conception to death. There are both natural fields, such as the Earth’s magnetic field, and those created by humans, such as in nuclear magnetic resonance tomography. The magnitude of the magnetic field is expressed as magnetic flux density B [Tesla (T) unit] or magnetic field strength H [ampere per meter (A/m) unit]. The Earth’s magnetic field reaches a value of 65 µT compared to 3.0 Tesla (=3,000,000 µT) in an NMR tomograph.

The biological effects of magnetic fields are poorly understood. The possible mechanisms of action and associated clinical applications have been discussed in detail by Markov [16]. An extremely low-frequency magnetic field (ELFMF) affects bioelectrical signals in the cerebral cortex [17] and has been used as a therapy for chronic progressive multiple sclerosis [18]. However, our understanding of the benefits of magnetic stimulation (MS) for patients with neurological disorders is incomplete (especially neurodegenerative diseases). The literature presents both positive and negative outcomes for this type of treatment [19]. There have been only a few studies on the clinical effects of ELFMFs published in well-known journals in the field [16]. Even after publication of the single report on the proven effects of transcranial MS (TMS) on altering oxy-Hb levels in the human brain cortex, there exists no clear scientific explanation as to how such responses were achieved [20], even though the intensity of the magnetic field used in TMS is several thousand times stronger than the intensity of ELFMFs. In recent years, alarming reports have emerged claiming that exposure to ELFMFs may lead to the development of ALS, an incurable, progressive motor neuron disease [21,22]. Therefore, studies in this area are essential to determine the benefit-risk balance of ELFMF therapy.

This pilot study aimed to evaluate the influence of MS on fluctuations in cortical blood oxygenation in the motor cortex using fNIRS. Gaining a better understanding of the mechanism(s) of MS functioning may impact the various treatment regimens currently used in patients with neurodegenerative diseases.

## 2. Materials and Methods

This study included 16 healthy, right-handed volunteers (8 females, 8 males) with a mean age of 27 (range 18–61).

We used an fNIRS apparatus to record cerebral near-infrared signals (NIRScout, NIRx Medical Technologies LLC, Glean Head, NY, USA). Lights from the eight dual-wavelength LED light sources (wavelengths 760 and 850 nm) were detected by eight optodes with avalanche photodiodes (APD) placed 3 cm from the emitters. The optodes and the channels (blue lines between emitters and detectors) were placed to cover the left and right motor cortices, as presented in Figure 1. The positioning of the completely mounted cap in a selected participant can be seen in Figure 2.

All measurements were recorded using the NIRStar 15.0 program (NIRx Medizintechnik GmbH, Berlin, Germany). Signal intensity was calibrated and verified for each channel before data collection, and the low pass filter (0.2 Hz) was used for data acquisition. In the NIRStar 15.0 program there are four quality classes of calibration: excellent, acceptable, critical, and lost. Only excellent level for all channels was accepted for every participant. In the case of insufficient strength of the signal, slight shift of the optode or optodes usually fixed the problem. Having successful calibration all scheduled tests were performed without additional recalibration. Participants were asked to remain supine on the magnetic stimulating mat (Viofor JPS, Med&Life Sp. z o.o., Komorów, Poland).

Four protocols were used to complete the research project. In the first protocol, Protocol I-A, the participants remained at rest with eyes closed during the length of the examination, according to the following scheme (the scheme is presented only to show relevant time points for data analysis): Rest 5 s → 5 × {Rest 10 s → Rest 20 s} → Rest 20 s. In protocol I-B, an ELFMF was emitted from the mat for 10 s with 20 s intervals between impulses. The estimated value of the magnetic induction was 11.5–276 µT. The impulses formed a multi-peak frequency spectrum from 0.08–195 Hz. The magnetic field covered the entire body, including the head and the brain. The timeline for protocol I-B was as follows: Rest 5 s → 5 × {Magnetic field 10 s → Rest 20 s} → Rest 20 s.

The second set of protocols included two tests (II-A and II-B). Again, participants were asked to stay supine on the magnetic stimulating mat with closed eyes. During the first test (II-A), the MS in the mat was off; motor cortex stimulation was achieved through a 10 s finger-tapping task performed simultaneously using both hands. The intervals between the stimulations lasted 20 s. This was followed by test II-B, in which the task in II-A was performed while the magnetic field was on. Parameters of data acquisition were the same as those used in the first protocol. Participants were blinded to the activation of the magnetic mat. The timeline for both protocols is presented below:

Protocol II-A:

Rest 5 s → 5 × {Finger-tapping 10 s → Rest 20 s} → Rest 20 s

Protocol II-B:

Rest 5 s → 5 × {Magnetic field + finger-tapping 10 s → Rest 20 s} → Rest 20 s

All data were analyzed using the MATLAB-based NIRSLab 15.0 program. Recordings were filtered with the bandpass filter, with a high cut-off frequency of 0.2 Hz and a low cut-off frequency of 0.01 Hz. The normal spectrum for oxy-Hb and deoxy-Hb used in the analyses was created according to the manufacturer’s recommendations. The final data for each participant was the mean of five replicates, covering 5 s before and 20 s after stimulation for each channel. Depending on the protocol used, data were averaged for left, right, and both hemispheres and they were visually and statistically evaluated after MS or finger-tapping.

Data on the maximum changes in oxy-Hb (oxy-Hbmax) concentrations, the time of occurrence of oxy-Hbmax (Tmax), and the difference between maximum and minimum changes in the concentration of oxy-Hb (Δoxy-Hb) were collected and analyzed.

To evaluate the influence of only the magnetic field on the change in the concentration of oxy-Hb in the motor cortex, the data obtained from protocol I-A were compared with those of protocol I-B (REST). To evaluate the magnetic field’s influence (continuous application) on oxy-Hb during finger-tapping, data from protocol II-A were compared to those from protocol II-B (MOTION). To prove the well-known effect of finger movements on the change in oxy-Hb concentration [23,24,25], protocol I-A was compared with protocol II-A (REST-MOTION).

### 2.1. Statistical Analysis

Data distribution was assessed by the Shapiro-Wilk test. For normally distributed data, the Student’s t-test was used; otherwise, the Mann-Whitney U-test was performed (Dell Inc. 2016. Dell Statistica [data analysis software system], version 13. Tulsa, USA software.dell.com) (accessed on 22 November 2021). Differences were considered statistically significant for *p*-values < 0.05.

### 2.2. Ethics Approval

The study was conducted in accordance with the local bioethical committee deci-sion, issued by the Pomeranian Medical University in Szczecin (approval reference num-ber KB-0012/77/18). All volunteers were informed of the purpose of the research and its process, and all participants provided written informed consent.

## 3. Results

We did not observe the characteristic blood oxygenation level-dependent responses in the left, right, or both hemispheres for any test in protocols I-A and I-B. As such, only Δoxy-Hb values [μmol/L] of participants with (MS) and without MS (no MS) were included in the statistical analysis (Table 1).

Non-normal distribution of data was observed in I-A and I-B groups, in contrast to normal distribution in II-A and II-B ones.

In the second protocol sessions, II-A and II-B, the characteristic blood oxygenation level-dependent responses, defined as the increase in the concentration of oxy-Hb and the decrease in deoxy-Hb levels, were observed in the cerebral motor cortex after the finger-tapping task, both with and without MS (Figure 3).

The statistical information on Δoxy-Hb when performing the finger-tapping task (MOTION) under MS or without MS are presented in Table 2.

The average time of maximum oxy-Hb (t_max_) for the motor cortex in the left, right, and bilaterally in protocol II-A was 8.7 s in all three measurements (min to max: 4.9–11.9 s; 5.0–12.2 s; 4.8–12.1 s). The corresponding data for protocol II-B were 8.2 s, 8.1 s, and 8.2 s (min to max: 3.7–14.7 s; 3.7–14.9 s; 3.7–14.9 s), respectively. There were no significant between-group differences for tmax nor the concentrations of oxy-Hbmax.

Results of statistical comparison for the groups of MOTION (finger-tapping task, with and without MS), REST (motionless patients, with and without MS), and REST-MOTION (motionless patients and for fingers tapping task) are presented in Table 3.

## 4. Discussion

Although we applied a very sophisticated method that was sensitive to changes in oxy-Hb levels as low as 10^−3^ mmol/L (1 μM) to measure fluctuations in motor cortex oxygenation levels, we did not observe any effects from the short series of MS (with the parameters set for this study) on the characteristic blood oxygenation level-dependent responses in this brain area. We found that ELFMFs alone had no measurable effect on the oxygenation levels of the brain motor cortex during rest or while performing finger-tapping, a motor task.

fNIRS is a state-of-the-art technique used to study cortical oxygenation. Despite decades of use, standard protocols for use of fNIRS in clinical or research settings have not been developed [13]. Consequently, research results obtained from fNIRS are difficult to compare between centres. Additionally, fNIRS has no place in routine clinical examinations. For example, changes in the concentration of oxyHb in the motor cortex following different hand movements varied significantly between studies, with values of 200 μM [4], 3 μM [25], and 0.8 μM [26] being reported. The latter resembled the values obtained in our study. In some previous studies, the units of measurements used were either arbitrary units or mol/(L·cm) instead of measuring the concentrations of oxy-Hb and deoxy-Hb and reporting any changes in these concentrations in μM. This was due to different processing and analysis algorithms with various values for differential pathlength factors and filter cut-off frequencies.

The capability of fNIRS to measure minute changes in the concentrations of oxy-Hb and deoxy-Hb (μM) attests to the quality and power of both the instrument and the study methods. Nonetheless, such measurements are prone to errors related to various—especially motion—artefacts. To avoid such errors, all tests were performed under strict conditions, ensuring no head movements, including those associated with saliva swallowing.

Due to our cohort’s small size, we did not divide the group by age and sex. Each participant had two research sessions with the study factor (I-B and II-B) and the corresponding control sessions (I-A and II-A). In session I-B, the magnetic field was both a stimulus and a test factor. In session II-B, the stimulus was the finger-tapping task, and the examined factor was the magnetic field. The effectiveness of the applied methodology was demonstrated by observing considerable changes in the concentration of oxy-Hb when comparing the values obtained during finger-tapping compared to those obtained at the resting state, which were statistically significant.

We did not observe any measurable effect of the magnetic field on the haemodynamic response of the motor cortex in either the passive (sessions I-A and B—REST) or the active (sessions II A and B—MOTION) experiments. Despite our findings, the impact of magnetic field biostimulation with different parameters and exposure times on the brain cannot be excluded. Future studies should examine the effects of magnetic fields with different parameters, such as a different time and intensity of exposure, on the oxygenation levels in the motor cortex and other cortical areas. The most important limitations of the study are small number of participants and not homogenous groups concerning age and gender, so it has to be treated as a pilot attempt.

In conclusion, we did not observe any measurable effect of the magnetic field on the level of oxyhaemoglobin in the motor cortex. The influence of MS on cortical blood oxygenation requires further research with larger cohorts to prove or negate any effect on the brain, bearing in mind possible harmful outcomes. Functional Near Infrared Spectroscopy is the only method to study the effect of a magnetic field on possible changes in hemoglobin levels in the brain vessels. Such studies could pave the way for additional, highly detailed studies on the use of this medical procedure in clinical practice.

## Figures and Tables

**Figure 1 ijerph-19-04012-f001:**
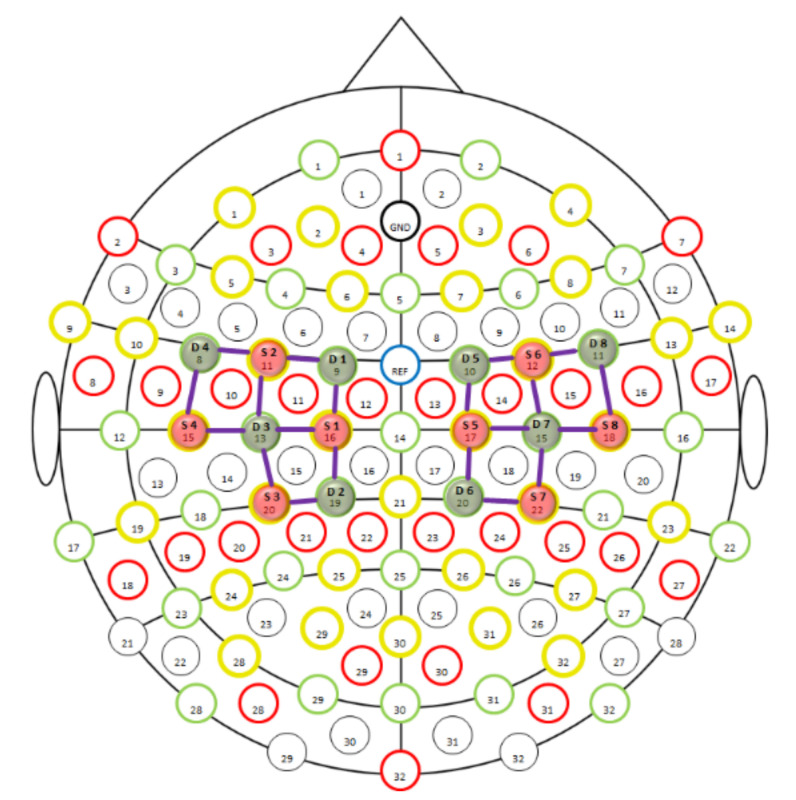
fNIRS montage, 8 × 8, for motor cortex; S, source; D, detector.

**Figure 2 ijerph-19-04012-f002:**
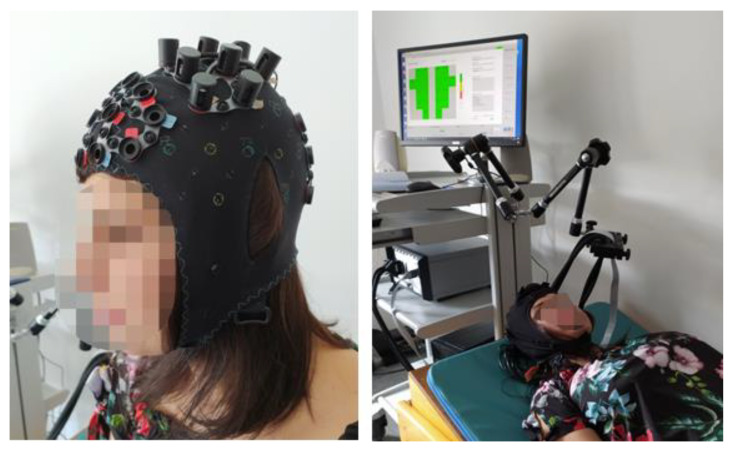
A person wearing the completely mounted cap.

**Figure 3 ijerph-19-04012-f003:**
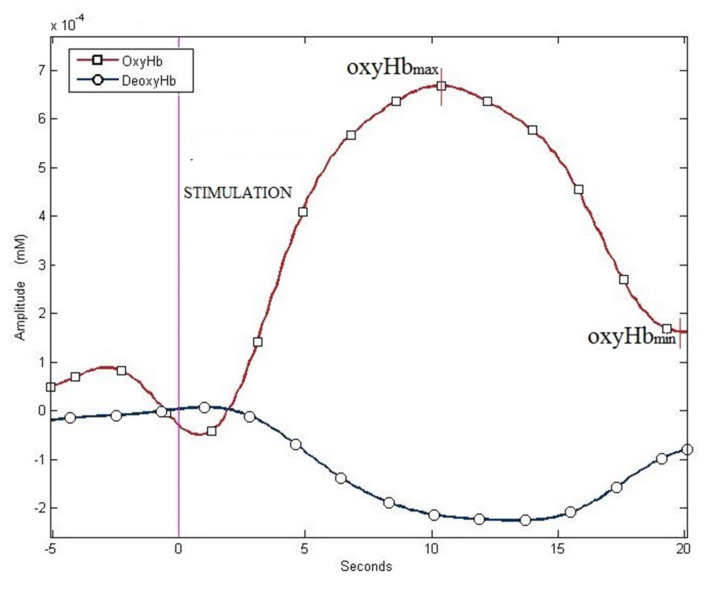
Examples of haemodynamic responses: increase in oxy-Hb and decrease in deoxy-Hb—data from protocol II-A.

**Table 1 ijerph-19-04012-t001:** Maximum and minimum changes in the concentration of oxyhaemoglobin in motionless participants, with and without magnetic stimulation.

REST		Δoxy-Hb [μmol/L]		
		Left Hemisphere	Right Hemisphere	Both Hemispheres
Mediana	no MS	0.21	0.20	0.21
	MS	0.17	0.20	0.18
Min	no MS	0.07	0.08	0.07
	MS	0.05	0.07	0.06
Max	no MS	0.59	0.63	0.61
	MS	0.41	0.30	0.32
25	no MS	0.12	0.13	0.11
	MS	0.08	0.10	0.09
75	no MS	0.27	0.26	0.28
	MS	0.34	0.27	0.31

**Table 2 ijerph-19-04012-t002:** Differences in the motor cortex oxygenation values after the finger-tapping task, with and without magnetic stimulation.

MOTION		Δoxy-Hb [μmol/L]		
		Left Hemisphere	Right Hemisphere	Both Hemispheres
Avarage	no MS	0.51	0.50	0.50
	MS	0.51	0.45	0.50
Min	no MS	0.21	0.19	0.20
	MS	0.14	0.14	0.14
Max	no MS	0.96	0.85	0.90
	MS	0.79	0.73	0.94
SD	no MS	0.21	0.21	0.21
	MS	0.21	0.21	0.24

**Table 3 ijerph-19-04012-t003:** Statistical comparison for the groups—MOTION [finger-tapping task, with and without magnetic stimulation], REST [motionless participants, with and without magnetic stimulation], and REST-MOTION [motionless patients (rest) and for finger-tapping task (motion) with and without magnetic stimulation].

	Hemisphere	*p*-Value			
		MOTION No MS vs. MS	REST No MS vs. MS	MS Rest vs. Motion	No MS Rest vs. Motion
**Δoxy-Hb**	Left	0.96	* 0.90	* <0.05	* <0.05
	Right	0.46	* 0.87	* <0.05	* <0.05
	Both	0.97	* 1.00	* <0.05	* <0.05

Differences between averages were considered statistically significant for *p*-values < 0.05. Student’s *t*-test was used for normally distributed data; otherwise, the * Mann-Whitney U-test was performed.

## Data Availability

The data used and/or analyzed during the current study are availablefrom the corresponding author on request.

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
