# Peer review of "Searching for the Mechanism of Action of Extremely Low Frequency Electromagnetic Field—The Pilot fNIRS Research"

_ijerph, 2022, doi:10.3390/ijerph19074012_

Round 1

Reviewer 1 Report

Dear Authors,

This paper is interesting and important due to the still pending problem of the benefits of magnetic stimulation in neurological disorders. The title is promising, but the content does not suit it as well as the second aim: „Gaining a better understanding of the mechanism(s) of MS functioning may impact the various treatment regimens currently used in patients with neurodegenerative diseases”. In my opinion the manuscript does not contribute to the recognition of the mechanisms.

I am afraid that the Conclusions resulting from this work may be misleading due to methodological flaws. The study protocol is carefully elaborated but , I believe that a study of such a small group would require the smallest possible age difference among the participants (range 5-10 years). Such a wide range of ages (18-61 yrs) can completely distort the result. The same is true for gender, as it is known that men and women react differently to different stimuli. Moreover, the statistical analysis raises doubts as the description shows that  probably  the methods to analyze independent samples were used.

Thus only the conclusion: ”The influence of MS on cortical blood oxygenation request requires further research with larger cohorts to prove or negate any effect on the brain, bearing in mind possible harmful outcomes” is convincing.

Reviewer 2 Report

In my opinion, this study is not complete. As the authors know, the U.S. Food and Drug Administration permitted marketing of the TMS for treatment of some disorders such as OCD. Although I agree that in the literature, both positive and negative outcomes for this type of treatment can be found, but to present a complete work and finally make a good conclusion, authors need to involve both healthy and impairments and do the experiments for a complete treatment period to make a better conclusion.

Round 2

Reviewer 1 Report

Dear Authors, thank you for changes you made in your manuscript. Although they do not solve my doubts, however, as a pilot study this manuscript work may be published.

Reviewer 2 Report

I still believe that for making a good conclusion, authors need to involve both healthy and impairments and do the experiments for a complete treatment period to make a better conclusion. It should be noted that not only OCD impairments can be involved in this study, but depression disorders and other disorders that can be treated by TMS can be studied in this work.

Author Response

This manuscript is a resubmission of an earlier submission. The following is a list of the peer review reports and author responses from that submission.